# Potential Treatment of Lysosomal Storage Disease through Modulation of the Mitochondrial—Lysosomal Axis

**DOI:** 10.3390/cells10020420

**Published:** 2021-02-17

**Authors:** Myeong Uk Kuk, Yun Haeng Lee, Jae Won Kim, Su Young Hwang, Joon Tae Park, Sang Chul Park

**Affiliations:** 1Division of Life Sciences, College of Life Sciences and Bioengineering, Incheon National University, Incheon 22012, Korea; muk365@inu.ac.kr (M.U.K.); licdbsgod@inu.ac.kr (Y.H.L.); topfkim@naver.com (J.W.K.); dasen43201@gmail.com (S.Y.H.); 2The Future Life & Society Research Center, Chonnam National University, Gwangju 61186, Korea

**Keywords:** lysosomal storage disease, mitochondrial–lysosomal axis, lysosome, mitochondria

## Abstract

Lysosomal storage disease (LSD) is an inherited metabolic disorder caused by enzyme deficiency in lysosomes. Some treatments for LSD can slow progression, but there are no effective treatments to restore the pathological phenotype to normal levels. Lysosomes and mitochondria interact with each other, and this crosstalk plays a role in the maintenance of cellular homeostasis. Deficiency of lysosome enzymes in LSD impairs the turnover of mitochondrial defects, leading to deterioration of the mitochondrial respiratory chain (MRC). Cells with MRC impairment are associated with reduced lysosomal calcium homeostasis, resulting in impaired autophagic and endolysosomal function. This malicious feedback loop between lysosomes and mitochondria exacerbates LSD. In this review, we assess the interactions between mitochondria and lysosomes and propose the mitochondrial–lysosomal axis as a research target to treat LSD. The importance of the mitochondrial–lysosomal axis has been systematically characterized in several studies, suggesting that proper regulation of this axis represents an important investigative guide for the development of therapeutics for LSD. Therefore, studying the mitochondrial–lysosomal axis will not only add knowledge of the essential physiological processes of LSD, but also provide new strategies for treatment of LSD.

## 1. Introduction

Lysosomes are membrane-bound organelles that contain a variety of digestive enzymes [1]. Lysosomes are involved in several cellular processes, including macromolecular catabolism and recycling [1]. Lysosomal storage disease (LSD) is a disorder caused by a deficiency of enzymes in lysosomes, leading to the accumulation of undigested or partially digested macromolecules [2,3]. LSD includes 50 rare monogenic disorders, classified according to the specific genetic defects and biochemical properties of the stored macromolecules [4,5,6,7,8,9]. LSD affects all parts of the body, including the skeleton, skin, heart, and central nervous system [10]. Recently, enzyme replacement therapy has been suggested as a treatment for LSD as it can alleviate symptoms [11,12,13]. However, this treatment has not been effective in restoring pathologic phenotypes to normal levels [11,12,13]. Although other clinical trials are ongoing for possible treatments for some of these diseases, there are currently no approved treatments for LSD.

Lysosomes and mitochondria have classical functions, namely as recycling bins and energy factories, respectively [14,15]. Mounting evidence supports functional interconnection and suggests it as a signaling platform to maintain cellular homeostasis [16]. Recently, a functional role of the mitochondrial–lysosomal axis in senescence alleviation has been proposed [14,17]. Proper control of this axis is essential for senescence alleviation, as evidenced by experiments showing that activation of this axis, by mitochondrial functional recovery, restored the senescent phenotypes [14,17]. Furthermore, the importance of this axis in senescence is emphasized by the fact that activation of this axis by lysosomal acidification has endowed an immortal germ lineage in *Caenorhabditis elegans* [18]. Thus, the modulation of this axis has been proposed as an important research guideline for the development of therapeutics for senescence [14,17].

The purpose of this review is to summarize LSD-related lysosomal/mitochondrial dysfunction and speculate on the crosstalk between these organelles. An extensive literature search in PubMed (https://pubmed.ncbi.nlm.nih.gov/) (accessed on 15 October 2020) was performed using search terms such as LSD-related lysosomal dysfunction, LSD-related mitochondrial dysfunction, mitochondrial–lysosomal crosstalk, and mitochondrial–lysosomal axis. In order to process and analyze the collected data and results, we used a systematic review strategy known as the gold standard for review writing in medical fields [19]. Based on the previous and recent findings that have been processed and analyzed (Appendix A), we will highlight the role of the mitochondrial–lysosomal axis in LSD and propose a crucial role for this axis as a regulator of LSD treatment.

## 2. Lysosomal Dysfunction in LSD

Lysosomes are membrane-bound organelles containing various lytic enzymes within the acidic vacuolar compartment [20,21]. They incorporate new hydrolytic enzymes by fusion with vesicles pinched off from the Golgi apparatus and break down a variety of substrates ranging from intracellular macromolecules to impaired organelles (Figure 1A; blue dot indicates functional lysosomal enzymes). Lysosomes fuse with intermediate vesicles, termed autophagosomes, where cargos are either sequestered in bulk (e.g., cytosols with all the contents in that area) or selectively (e.g., mitochondria, endoplasmic reticulum or ribosomes) [22]. Subsequent fusion of autophagosomes with lysosomes creates autolysosomes, where lysosomal enzymes degrade the autophagosome-delivered substrates (Figure 1A). Extracellular substances are internalized and delivered from the plasma membrane to the endosome (Figure 1A). Lysosomes fuse with endosomes to form highly dynamic membrane structures called endolysosomes, in which endocytosed substrates are degraded and recycled (Figure 1A). The digestion of autophagocytosed or endocytosed substrates occurs in an acidic environment of the lysosomal lumen (pH 4.5 to 5.0), which is maintained by a vacuolar ATPase (V-ATPase) proton pump [23,24]. V-ATPase uses the energy of ATP hydrolysis to move protons (H^+^) from the cytoplasm to the lysosome lumen against a concentration gradient (Figure 1A). In addition, the digestion of substrates occurs at an appropriate level of calcium ions, which are maintained by mammalian mucolipin TRP (Transient Receptor Potential) channel subfamily (*TRPML1*) transporters in the lysosomal membrane [25]. Lysosomal Ca^2+^ homeostasis plays an important role in the regulation of lysosomal function, including vesicular trafficking and fusion processes [26].

LSD is a metabolic disorder caused by mutations in genes that encode lysosomal enzymes such as lysosomal glycosidases, proteases, enzyme modifiers or activators [27,28]. Mutations alter the functional activity of lysosomal enzymes, leading to abnormal degradation of substrates and accumulation of various substrates inside the lysosome (i.e., “storage”) (Figure 1B; red dot indicates dysfunctional lysosomal enzymes). These events exacerbate the activity of lysosomal enzymes that are not genetically deficient (Figure 1B). Lysosomes loaded with undigested substances exhibit defective fusion with autophagosomes, leading to the accumulation of unfused autophagosomes (Figure 1B). For example, LSD cells have an increased number of organelles containing autophagosome markers, but only a few of them display lysosomal markers, suggesting defective fusion between lysosomes and autophagosomes [29]. Furthermore, lysosomes of Niemann–Pick type A/C disease, a type of LSD, exhibit alterations in endolysosome trafficking characterized by the accumulation of extracellular substances and specific lipid species in lysosomes [30]. The altered lysosomal pH represents another characteristic of LSD [31,32]. In LSD, lysosomes loaded with undigested lipids with free amine groups in cationic lipids cause severe lysosomal and cellular dysfunction, including abnormal alkalization of lysosomal pH [31]. The altered lysosomal pH subsequently deteriorates the activity of various lysosomal enzymes (Figure 1B). In addition, dysregulation of lysosomal Ca^2+^ homeostasis constitutes another feature of LSD (Figure 1B). For example, one type of LSD, Mucolipidosis type IV (MLIV), is caused by a loss-of-function mutation in *TRPML1* associated with lysosome-mediated Ca^2+^ efflux [33]. The decreased Ca^2+^ release via *TRPML1* disrupts endocytosis and vesicular fusion, which limits the access of lipids to lysosomal lipases, along with the accumulation of undegraded lipids in MLIV [33].

## 3. Mitochondrial Dysfunction in LSD

Mitochondria are double-membrane organelles that produce most of the cellular ATP, which is used as a source of a cell’s biochemical reaction [34,35]. These highly dynamic organelles can be processed through successive fusion and fission cycles (Figure 2A). Cytoplasmic Ca^2+^ modulates GTPase activity, which regulates the balance of mitochondrial fusion/fission events (Figure 2A). The optimal Ca^2+^ concentration in the cytoplasm is primarily controlled by lysosomal Ca^2+^ efflux through *TRPML1* and Ca^2+^ release through the endoplasmic reticulum (ER) (Figure 2A) [36]. Although mitochondrial fusion contributes to a relatively homogeneous network by mixing the contents of partially damaged mitochondria, mitochondrial fission is required to generate new mitochondria by eliminating damaged mitochondria through autophagy [37]. Functional lysosomes are essential for autophagy-mediated removal of defective mitochondria (mitophagy; Figure 2A). Specifically, mitophagy promotes turnover of mitochondria and prevents the accumulation of dysfunctional mitochondria [38].

Abnormal mitochondrial morphology is a pathogenic feature of LSD [39,40]. Dysregulation of lysosomal Ca^2+^ homeostasis in LSD interferes with the balance of mitochondrial fission and fusion cycles, leading to mitochondrial morphology changes, namely mitochondrial fragmentation or elongation (Figure 2B). These abnormalities are associated with degenerative mitochondrial functions, such as qualitative changes in mitochondrial membrane potential (ΔΨm) and MRC function [41,42]. Support for this phenomenon is evident from observations showing that a mouse model of LSD, produced by knocking out sulfatase-modifying factor 1, exhibits mitochondrial fragmentation accompanying impaired ATP production [43]. Moreover, fragmented mitochondria with reduced ΔΨm are present in neurons and astrocytes cultured from another LSD mouse model, GM1 gangliosidosis [44]. Furthermore, the elongated mitochondrial morphology as a characteristic in LSD can be inferred by evidence showing the accumulation of swollen and elongated mitochondria with disorganized crista in LSD cells [45]. The elongated mitochondria also exhibit reduced MRC function concomitant with decreased ATP production. Collectively, these observations show that abnormal mitochondrial morphology, primarily attributed to the perturbation of Ca^2+^ homeostasis, constitutes one of the salient features observed in LSD.

Accumulation of dysfunctional mitochondria is a surrogate marker for abnormal mitochondrial morphology in LSD. The accumulation of impaired mitochondria is primarily because of lysosomal deficiency in LSD [42]. In particular, lysosomal deficiency deteriorates Ca^2+^ buffering, consequently inhibiting the sequential stages of pre-autophagosome formation and autophagosome maturation (Figure 2B). Even after lysosomal fusion with autophagosomes, inactive hydrolytic enzymes in lysosomes are unable to degrade dysfunctional mitochondria (Figure 2B). For example, in LSD because of alpha-glucosidase deficiency, dysfunctional mitochondria are sequestered in autophagy vesicles, but not digested by lysosomes, suggesting inefficient lysosomal-autophagy-mediated degradation (Figure 2B) [46]. Support for this finding is evident in previous results showing that the accumulation of mitochondria with various abnormalities was observed in LSD, including MLII, MLIII [47], MLIV [48], GM1 gangliosidosis [44,49], or Batten disease [50]. Dysfunctional mitochondria are not efficiently eliminated and constitute a major source of reactive oxygen species (ROS) production. Excessive ROS causes severe damage to lysosomes, exacerbating the accumulation of damaged mitochondria [51]. The accumulation of dysfunctional mitochondria is a well-known feature of LSD and leads to deleterious consequences, including altered cellular homeostasis and tissue degeneration [52].

## 4. Importance of the Mitochondrial–Lysosomal Axis in LSD

Lysosomes and mitochondria interact with each other, and this crosstalk can act as a central hub at various levels [14,16]. They constantly inform each other of their functional status through retrograde or anterograde signals [53]. These organelles are recognized as signaling platforms that regulate many key elements of cellular physiology and are more than simple units, with interconnections through the mitochondrial–lysosomal axis [16]. 

The effect of lysosomal function on mitochondria is systematically characterized, indicating that the regulatory mechanisms of lysosomes control mitochondrial function. As mentioned in Section 3, lysosomal defects in LSD impair the mechanism by which the damaged mitochondria are removed (Figure 3A). Thus, damaged mitochondria accumulate concomitant declines in mitochondrial function, such as decreased ATP production and increased ROS generation (Figure 3A). In most cases, lysosomes regulate mitochondrial function without physical contact, but sometimes lysosomes control mitochondrial function through interorganelle contacts [54,55]. For example, Ca^2+^ transport is mediated by the lysosomal channel TRPML1 in lysosome–mitochondrial contact [56]. Thus, a deficiency of *TRPML1* function in MLIV results in a decrease in Ca^2+^ concentration at the lysosomal–mitochondrial interface [56]. In turn, the decrease in Ca^2+^ concentration reduces contact-dependent mitochondrial calcium absorption by the mitochondrial calcium uniporter (MCU), impairing calcium-dependent mitochondrial function, including oxidative phosphorylation and ATP production [33,57,58]. Therefore, these findings support a potential role for lysosome in the regulation of mitochondria, in a contact-independent or dependent manner.

The reciprocal crosstalk between mitochondria and lysosomes suggests that the regulatory mechanisms of mitochondrial function are essential for the proper functioning of lysosomes. For example, a mouse model with a genetic deletion of mitochondrial transcription factor A (*Tfam*) was generated, and the effects of mitochondria on lysosomes were evaluated [59]. *Tfam* encodes a mitochondrial transcription factor that plays an essential role in the transcription and replication of the mitochondrial genome [60,61]. *Tfam*-deficient cells show abnormal levels of mitochondrial DNA expression concomitant with reduced MRC function and altered mitochondrial metabolism (Figure 3B). Cells with reduced MRC function are associated with reduced lysosomal Ca^2+^ efflux, resulting in impaired endolysosomal function with an abnormal accumulation of lipid species, including sphingomyelin (Figure 3B). Extending the relevance of these findings, the deficiency of the *Tfam* deteriorates mitochondrial function, which was accompanied by disturbances in endolysosomal trafficking and autophagy function [59]. Thus, these findings suggest that there is an effect of mitochondria on the regulation of lysosomal function, which underlies the operation of the mitochondrial–lysosomal axis.

The functional interconnection between lysosomes and mitochondria is highlighted by the fact that lysosomal acidification is achieved by V-ATPases, which require energy in the form of ATP energies, which are provided by the functional mitochondria [62,63]. As noted in Section 2, V-ATPase contributes to maintaining an acidic environment in the lysosomal lumen, allowing most of the lysosomal enzymes to be active (Figure 3C). However, reduction of MRC function in LSD prevents mitochondria from supplying sufficient ATP to V-ATPase for proper functioning in the lysosome membrane [64,65]. This finding has been established in recent studies showing that inhibition of mitochondrial function after deletion of the mitochondrial proteins (AIF, OPA1 or PINK1) reduces the activity of MRC, thereby reducing ATP production [66]. Furthermore, in cells with reduced MRC function, lysosomal pH did not change even after suppressing V-ATPase activity with bafilomycin, but in normal cells, lysosomal pH changed after suppressing V-ATPase [66]. These results indicate that functional mitochondria are necessary for V-ATPase-mediated lysosomal acidification. Thus, the functional interdependence between these organelles suggests the role of the mitochondrial–lysosomal axis as a promising and potent target for LSD treatment.

## 5. Strategies to Activate the Mitochondrial–Lysosomal Axis in LSD

Mitochondria and lysosomes constantly interact with each other in a reciprocal manner. Functional interconnection via the mitochondrial–lysosomal axis suggests that the regulatory mechanisms of one organelle are essential for the proper functioning of another organelle [14,16]. Therefore, it is appropriate to assume that proper control of the mitochondrial–lysosomal axis might represent a new therapeutic frontier in LSD treatment.

Recent studies have demonstrated that the pathophysiologic symptoms of LSD could be restored to a normal state by activation of the mitochondrial–lysosomal axis [67,68,69]. Specifically, Gaucher’s disease (GD), caused by a defect in lysosome β-glucocerebrosidase (GCase), exhibits impaired electron transport in MRC coupled with increased mitochondrial ROS generation and decreased ATP production [70]. In accordance with this finding, GD cells have a significantly lower content of coenzyme Q10 (CoQ), which is known to enhance MRC activity by accepting one electron from complexes I/II and transferring the reducing equivalent to complex III (Figure 4A; Green CoQ indicates a low content of CoQ). Given that mitochondrial MRC deficiency plays a crucial role in the pathogenesis of GD, GD cells have been supplemented with CoQ [71]. CoQ supplementation results in an improvement in mitochondrial function, indicated by a marked increase in cellular ATP levels and a marked decrease in ROS generation (Figure 4A; Pink CoQ indicates a high content of CoQ). Restoration of mitochondrial function ameliorates GCase defects in GD, resulting in reduced glycosaminoglycan accumulation [70]. The significance of mitochondrial functional recovery on LSD treatment is further supported by experiments with NAD^+^, which acts as an electron carrier in MRC during oxidative phosphorylation [72]. Indeed, increasing intracellular levels of NAD^+^ led to marked improvement of the MRC defect [67,68,69]. Specifically, in mice lacking acid sphingomyelinase (ASM) activity, NAD^+^ treatment enhances ASM activity, reducing substrate accumulation in lysosomes [59]. These findings suggest that a strategy to activate the mitochondrial–lysosomal axis through restoration of mitochondrial function might be effective in treating LSD.

The mitochondrial–lysosomal axis is not unidirectional but functions bidirectionally to treat LSDs. For example, activation of autophagy with trehalose, an autophagy inducer, improves pathological symptoms in a mouse model of mucopolysaccharide IIIB, a type of LSD [73]. Specifically, trehalose treatment efficiently clears autophagy vacuoles from neurons, along with activation of the transcription factor EB (TFEB), which regulates lysosome biogenesis (Figure 4B). Furthermore, improvement of LSD symptoms by trehalose was blocked by *Atg7* knockdown, indicating that autophagy activation is a prerequisite for treating neurological symptoms associated with LSD [73]. Similar to this observation, promoting autophagy maturation and subsequent exocytosis in LSD enables the rescue of pathological storage and restoration of normal cellular state [74]. These data suggest that activation of the mitochondrial–lysosomal axis through activation of autophagy might be an alternative strategy for LSD treatment.

The regulation of the mitochondrial–lysosomal axis via lysosomal acidification represents a promising potent strategy for LSD treatment. Almost all LSDs have shown defects in lysosomal acidification, which provides an optimal environment for the degradation of macromolecules delivered to the lysosome and facilitates autophagic flux [30,75]. Lysosomal acidification is regulated by V-ATPase, composed of a cytoplasmic V_1_ sector and a lysosomal membrane-bound V_0_ sector. The reversible assembly of the V_1_/V_0_ sectors induces a functionally active V-ATPase to maintain lysosomal acidification [21,76]. A recent study revealed that V_1_/V_0_ assembly of V-ATPase is regulated by the inhibition of Ataxia telangiectasia mutated (*ATM*) gene [77]. Specifically, attenuation of ATM activity promotes V_1_/V_0_ assembly in V-ATPase accompanied by re-acidification of lysosomes (Figure 4C). Lysosomal re-acidification restores the activity of lysosomal enzymes and increases autophagic flux, which in turn accelerates the elimination of dysfunctional mitochondria with restoration of mitochondrial function [77]. Thus, activation of the mitochondrial–lysosomal axis through lysosomal re-acidification may enable an optimal environment for lysosomal enzymes, providing therapeutic benefits in situations of lysosomal dysfunction. 

## 6. Conclusions and Perspectives

Functional interconnection between lysosomes and mitochondria has been reviewed in several review articles. One review specifically addressed the coordination between mitochondria and lysosome with a focus on cellular metabolism and signaling [53]. The importance of reciprocal crosstalk was discussed but was limited to its role in neurodegeneration [53]. This limitation of relevance was improved by another review addressing the potential role of mitochondrial dysfunction in the pathophysiology of LSD [42]. This review suggested that mitochondrial damage can lead to lysosomal dysfunction, supporting a common signaling pathway and crosstalk between the two organelles [42]. Although the causes and consequences of mitochondrial damage have been extensively described, no specific strategy to induce mitochondrial function recovery has been proposed [42]. Most recently, a review paper has been published focusing on mitochondrial dysfunction as an important contributing factor in the pathophysiology of LSD [78]. This review explained that dysfunctional mitochondria affect the function of lysosomes by generating ROS or depriving lysosomes of ATP [78]. Thus, a putative mechanism capable of causing mitochondrial dysfunction in LSD has been proposed as a therapeutically effective target in patients with LSD [78]. However, in LSD, many cellular pathways are impaired, so it remains unclear whether treatment strategies aimed at improving only mitochondrial dysfunction will be effective in the treatment of these disorders.

In this review, we summarized LSD-related lysosomal/mitochondrial dysfunction and investigated the crosstalk between lysosomes and mitochondria. Functional interconnections between lysosomes and mitochondria suggest that the regulatory mechanisms of one organelle are required for the proper functioning of another organelle. Thus, we proposed the mitochondrial–lysosomal axis as a potential therapeutic target for treating LSD (Figure 5). Regulation of this axis is bidirectional, either through restoration of mitochondrial function or restoration of lysosome function. Proper regulation of this axis, which is not dependent on one approach, may rescue the pathological symptoms of LSD, where many cellular pathways are impaired (Figure 5). A deeper understanding of the molecular mechanisms supporting the role of this axis in the onset and progression of LSD will pave the way for effective treatment of patients with LSD.

## Figures and Tables

**Figure 1 cells-10-00420-f001:**
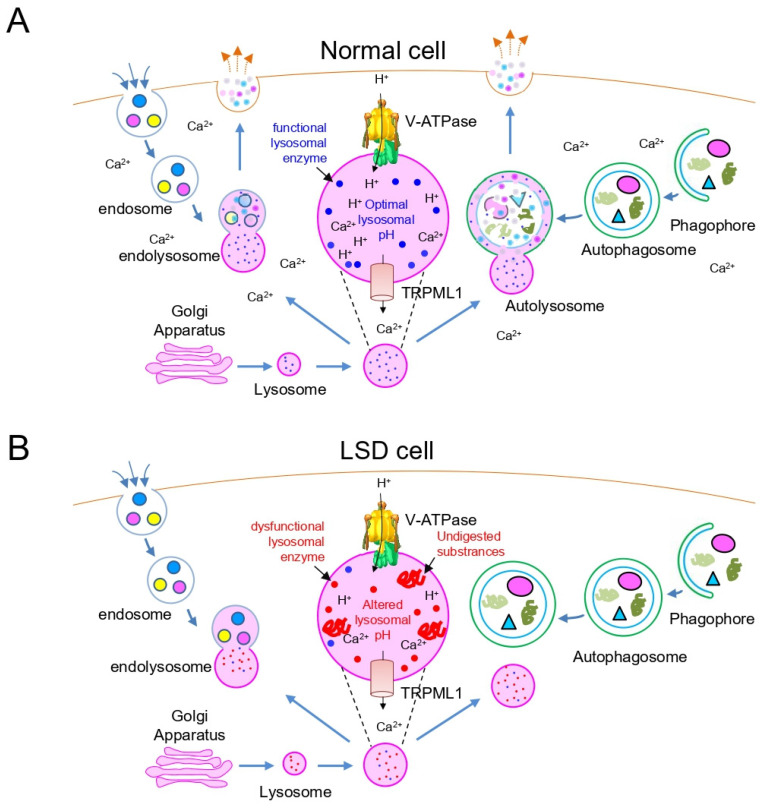
Schematic representation of the lysosome and autophagy pathway in normal cells (**A**) and lysosomal storage disease (LSD) cells (**B**). (**A**) Lysosomes fuse with autophagosomes to form autolysosomes. Lysosomes also fuse with endosomes to form highly dynamic membrane structures called endolysosomes. Functional lysosomes require an acidic environment in the lysosomal lumen (pH 4.5 to 5.0) maintained by the V-ATPase proton pump and an adequate level of calcium ions maintained by the TRPML1 transporter. TRPML1: mammalian mucolipin TRP channel subfamily, V-ATPase: a vacuolar ATPase. (**B**) LSD is a metabolic disorder caused by mutations in genes that encode lysosomal enzymes, consequently leading to accumulation of various substrates. Lysosomes loaded with undigested substances lead to defective lysosomal fusion with autophagosomes or endosomes. Furthermore, lysosomes of LSD exhibit alterations in lysosomal pH and lysosomal Ca^2+^ homeostasis.

**Figure 2 cells-10-00420-f002:**
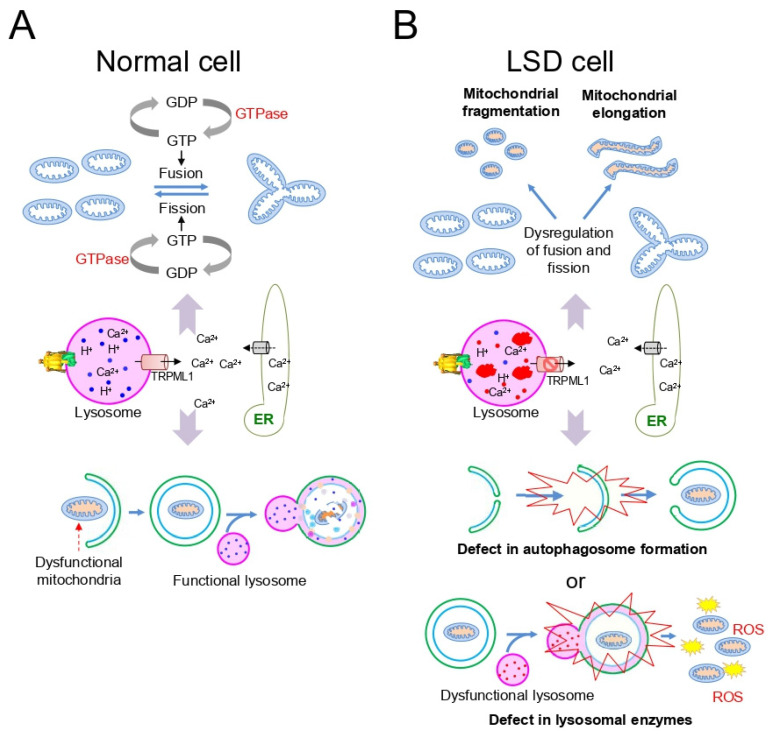
Schematic representation of the basic mechanisms of mitochondrial homeostasis in normal cells (**A**) and LSD-related mitochondrial dysfunction in LSD cells (**B**). (**A**) Cytoplasmic Ca^2+^ modulates GTPase activity, which regulates the balance of mitochondrial fusion/fission events. ER: endoplasmic reticulum. (**B**) Dysregulation of lysosomal Ca^2+^ homeostasis in LSD interferes with the balance of mitochondrial fission and fusion cycles. Lysosomal deficiency deteriorates Ca^2+^ buffering.

**Figure 3 cells-10-00420-f003:**
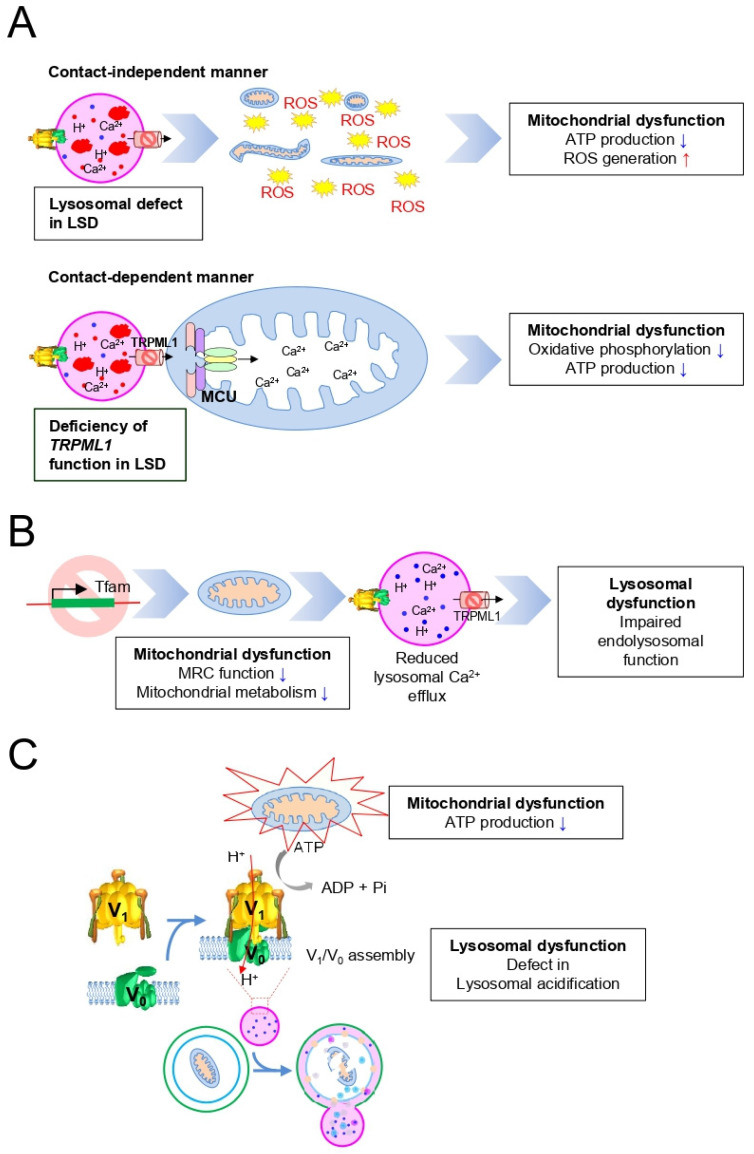
Importance of the mitochondrial–lysosomal axis in LSD. (**A**) Schematic representation of the basic mechanisms describing the effect of lysosomal function on mitochondria, in a contact-independent or dependent manner. MCU: mitochondrial calcium uniporter. (**B**) Schematic representation of the underlying mechanisms explaining the deletion effect of the mitochondrial transcription factor *Tfam* on lysosomal function. *Tfam*: mitochondrial transcription factor A. (**C**) Schematic representation of the underlying mechanisms explaining the effect of low ATP production due to mitochondrial dysfunction on lysosomal acidification.

**Figure 4 cells-10-00420-f004:**
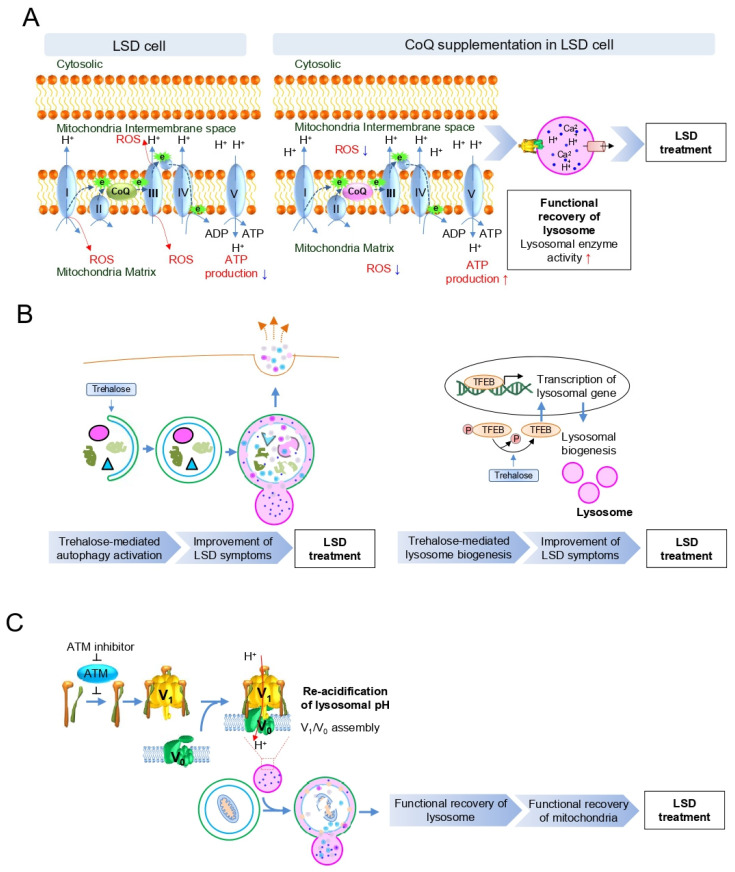
Strategies to activate the mitochondrial–lysosomal axis in LSD. (**A**) The strategy of activating the mitochondrial–lysosomal axis through coenzyme Q10 (CoQ) supplementation improves mitochondrial function to restore lysosome function. Green and pink CoQ indicate low and high CoQ content, respectively. ROS: reactive oxygen species. (**B**) The strategy of activating the mitochondrial–lysosomal axis through autophagy activation with trehalose improves pathological symptoms associated with LSD. (**C**) The strategy of activating the mitochondrial–lysosomal axis through lysosomal re-acidification restores the activity of lysosomal enzymes and increases autophagic flux with restoration of mitochondrial function.

**Figure 5 cells-10-00420-f005:**
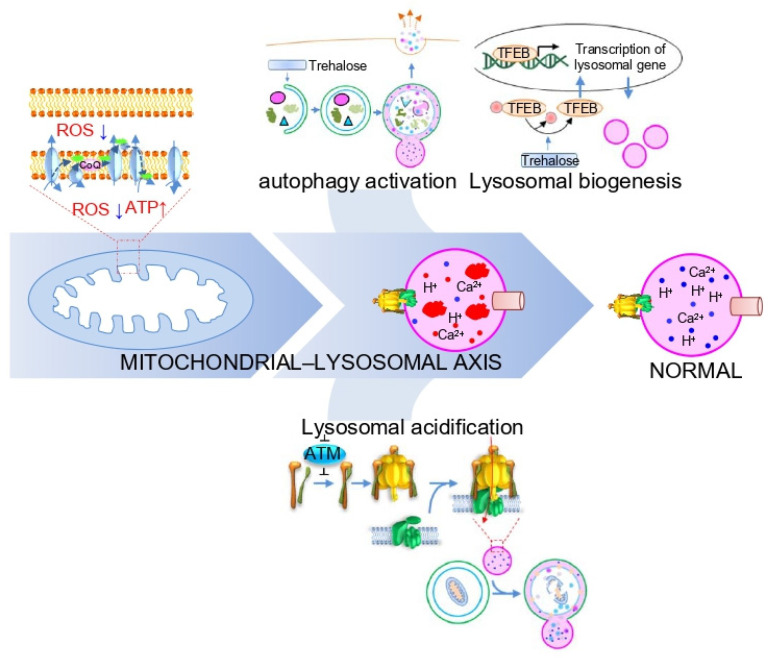
Synoptic representation of the mitochondrial–lysosomal axis contributing to the control of LSD. The activation of the mitochondrial–lysosomal axis through mitochondrial functional recovery or lysosomal functional recovery represents a promising and potent strategy to treat LSD.

## Data Availability

No new data were created or analyzed in this study. Data sharing is not applicable to this article.

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
