# Peer review of "Potential Treatment of Lysosomal Storage Disease through Modulation of the Mitochondrial—Lysosomal Axis"

_cells, 2021, doi:10.3390/cells10020420_

Round 1
Reviewer 1 Report
While the paper presents an interesting topic, I have some major comments and concerns that need to be addressed and clarified before further consideration.
Major concerns
- This topic has been extensively reviewed and or partially documented in several recently published review articles. The paper in its current form does not provide any novelty. This point should be highly considered and should be clarified and addressed, otherwise, the paper will not provide any significance. The authors should provide the novelty that has been added to this paper in comparison with the recently published review articles, which have covered the same topic.
- The paper contains only 66 references, reflecting that the authors have made a superficial literature search or just repeated what has been recently published.
- The title should be modified as Potential Treatment of Lysosomal Storage Disease Through Modulation of the Mitochondrial–Lysosomal Axis. Since the mechanism via modulation of the mitochondrial-lysosomal axis has not been validated clinically, so it should be mentioned as ‘’potential treatment’’.
- It would be better to add information about the period of published studies covered by this review article. This is to ensure that the paper is up to date.
- Another section named, Methodology would be useful to be included in the paper to provide the readers with information about what type of strategies/methods were used or followed to ensure the quality of processing the collected data and the outcomes. Also, information about the used databases for collecting and or extracting the data. Alternatively, this information could be highlighted in the Introduction section.
- Figures 1,2,3, and 4. All descriptions below these figures are too long. I recommend the authors describe the data presented in all figures in a specific way. Detailed descriptions could be discussed in the corresponding sections.
Minor concerns
Ca2+. This should be written as Ca2+. I recommend the author check such typing errors throughout the full text.
Since the manuscript is only a repetition of previously reviewed data from already published comprehensive reviews, I could not find a special characteristic for the current review; rather it was a repetition of previous works with superficial and general knowledge. I would encourage the authors to address this topic from a different angle.
Reviewer 2 Report
The authors have summarized literature data on possible interactions between mitochondria and lysosomes, and proposed strategies to modulate the mitochondrial-lysosomal axis in lysosomal storage disease.
The review is interesting; however, there are several points that can not be neglected.
First of all, while reading the manuscript I have had a "somewhere already seen - deja vu" feeling. And this is true, as a chosen way to present the story, the style of the manuscript, and even the sentences seem very similar to the paper published by the same team in 2018 (Ref. 11): compare the Chapter 2
(lines 57-66), the Chapter 4 (lines 166-179), the Chapter 6 (the Fig. 5 and lines 306-314). Therefore, could you, please, check the text of your manuscript for possible similarities to published papers?
In addition, almost every sentence contains a reference to a certain scientific paper, e.g., the Chapter 3, and this is true for all other parts. I suggest to rethink the way of presentation.
Moreover, the Fig. 5 looks too naive, and it does not provide any useful information. I propose using the Figure as a graphical abstract.
I also think that the authors could pay their attention to a recently published review on the same topic: doi.10.1016/j.molmed.2019.10.009.
And finally, there are several places that need to be corrected or clarified: brain is a part of CNS (see the lines 35-36); "mutations alters(?)" (line 91); lines 80-107 - do they belong to the Fig. 1? A Figure legend (line 123) - a certain part of the text is missing.
Round 2
Reviewer 1 Report
The manuscript has been significantly improved.
Reviewer 2 Report
Thank you, the authors have addressed all my concerns.